# Distance-Dependent Migration Intention of Villagers: Comparative Study of Peri-Urban and Remote Villages in Indonesia

Ar. Rohman T. Hidayat [1,2,*] , Kenichiro Onitsuka [3] , Corinthias P. M. Sianipar [4,*] and Satoshi Hoshino [3]

1   Graduate School of Global Environmental Studies (GSGES), Kyoto University, Kyoto 606-8501, Japan
2   Department of Regional and Urban Planning, Brawijaya University, Malang 65145, Indonesia
3   Department of Global Ecology, Kyoto University, Kyoto 606-8501, Japan; onitsuka.kenichiro.8m@kyoto-u.ac.jp (K.O.); hoshino.satoshi.5m@kyoto-u.ac.jp (S.H.)
4   Division of Environmental Science and Technology, Kyoto University, Kyoto 606-8502, Japan
*   Correspondence: a.r.taufiq.h@ub.ac.id (A.R.T.H.); iam@cpmsianipar.com (C.P.M.S.)

**Abstract:** Rural-to-urban migration disturbs essential factors of rural development, including labor forces, land ownership, and food production. To avoid late responses to emigration, scholars have begun investigating earlier stages of rural emigration. However, prior studies have focused on a single spatial entity only while also leaning toward trends in developed countries. Therefore, this study fills gaps by focusing on the differences in migration intention between villages in less developed settings. In observing the differences, this research takes peri-urban and remote villages as cases located at different distances from their nearest urban destination. This study treats migration intention as the dependent variable while using single-indicator place attachment and multi-indicator information sources as the independent variables. This work applies the Mann–Whitney U, ANOVA, and Brown–Forsythe tests on three hypotheses. This research also uses SEM-PLS to investigate the correlation model of the observed variables for each case. The results show that information sources negatively affect migration intentions in peri-urban settings. Remote rural areas also show similar results for the information sources variable; however, place attachment in remote settings significantly contributes to migration intention. These results show that place attachment and information sources contribute differently, depending on the distance to the urban area. We argue that access to public services and infrastructure contributes to the results. The findings suggest that an increased availability of information sources impedes the formation of migration intentions. Thus, this study suggests the necessity of improving rural infrastructure and public services to improve information literacy. It helps the government control rural emigration while fulfilling its obligation for rural development. It also offers better rural livelihoods during the development progress, providing economic incentives for villagers to stay in villages.

**Keywords:** rural migration; urbanization; villagers; peri-urban village; remote village; developing country; rural youth; intergenerational gap; infrastructure development; governmental role

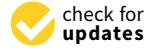



## 1. Introduction

The emigration of rural residents may occur temporarily or permanently, delivering positive and/or negative impacts on their rural origins (Mendola 2010). Various drivers stimulate the emigration of villagers, which may include factors relevant to their economic situation (Cheng et al. 2006; Lyu et al. 2019; Marta et al. 2020), life satisfaction (Liu and Pan 2020), natural disasters (Berlemann and Steinhardt 2017; Gray and Mueller 2012; Ishtiaque and Nazem 2017), and education (Crivello 2011). Besides internal reasoning, as such, there are records of rural residents who emigrate due to joining government-arranged projects such as the transmigration program (Abdoellah 1987). Considering the vast possibilities of drivers and impact, research on rural emigration have taken different angles, including

environmental (see Kelley et al. 2020), personal (Ayhan et al. 2020), community (Ajaero and Onokala 2013; de Brauw 2019), regional (Cohen 2006; Fasbender 1989), national (Yüksel et al. 2018), or international perspectives (Lacroix et al. 2016; van Dalen and Henkens 2012; Woods 2016). In the literature, Carling and Collins (2018) indicated that studies on migration drivers have been increasingly attracting attention from numerous scholars. In these studies, the intention to migrate has emerged as a critical variable that affects migration in general. Technically, the formation of a migration intention is an early step of the migration, which presumably contributes to actual migration (Abdelwahed et al. 2020). In keeping up with the emerging trend, various research has been advocating for the investigation of the intention to migrate among villagers. Of the literature that focused on migration intention, some have attempted to address the build-up process (Moon et al. 2010; Perez-Barbosa and Zhang 2017; Ramos 2019; Seyfrit et al. 2010; Traikova et al. 2018; Wolfe et al. 2020), relevant policies (Dufhues et al. 2020), and the link to actual migration (Tjaden et al. 2019). In general, scholars investigate the migration process from personal points of view. During the development of migration intentions, scholars have suggested multidimensional aspects that simultaneously affect the build-up process of the migration intention. These aspects include place attachment, social networks, and Internet use (Hiwatari 2016; Meyer 2018; Moon et al. 2010; Thulin and Vilhelmson 2016; Wolfe et al. 2020).

In the literature, several studies have discovered that attachment to the place of origin contributes to migration (Njwambe et al. 2019; Pretty et al. 2006). As an example, research focusing on rural Indonesia has discovered that a lower intention of villagers to stay in the village results in a lower place attachment than those with a stronger intention to stay (Priatama et al. 2019). Considering the urban area as a migration destination, other research focusing on a developed country has shown that villagers who live further from urban areas tend to have a stronger place attachment than those living at a closer distance (Gieling et al. 2019). Geographical variation has been indicated as an influencing force to place attachment. In addition to place attachment, distances between the places of origin and prospective destinations delay information flow, making distance an intervening obstacle to the development of migration intention. In the contemporary information era, information and communication technology (ICT) has made the flow of information between distant geographical locations faster, better, and arguably more affordable (Blank et al. 2018; Brunn 1998; Castells 2002; Pamungkas 2017). Consequently, advances in ICT have been reducing distance-related hurdles to migration. In particular, the Internet has been suggested to positively impact the development of migration intention among villagers by facilitating its users to seek information on prospective destinations throughout the world (Moon et al. 2010). It arguably accumulates information along with the build-up process of the intention to migrate, suggesting Internet-induced information gathering to positively or negatively affect migration intention (Moon et al. 2010; Onitsuka and Hidayat 2019; Vilhelmson and Thulin 2013). Since the Internet also delivers an accessible communication medium, it enables its users to expand their social network virtually (Laniado et al. 2018; Priatama et al. 2019). Technically, virtual communication allows prospective migrants to connect with other users who have similar interests, which, given the applicable algorithm of social network platforms (Li et al. 2019), could include those seeking information on similar prospective destinations. Furthermore, it would open possibilities for prospective migrants to connect with people living in prospective destinations (Dekker et al. 2016; Dekker and Engbersen 2014), who possibly are active migrants themselves. The accumulation of passive and active information sources occurs alongside an internal contemplation of migration intention. In short, there are compounded influences of Internet use on rural emigration (Hiwatari 2016).

However, the benefits of using the Internet and the expanded online social network do not necessarily result in an entirely diminished influence of physical distance (Laniado et al. 2018). In fact, any physical rural migration would go through physical distance in order to occur, underlining the critical considerations required over location and distance

during the build-up process of migration intention (Correa and Pavez 2016; de Groot et al. 2011; Docquier et al. 2014). In the literature, however, an extremely limited number of studies have focused on distance as an influencing factor during the build-up process of migration intention among villagers. Docquier et al. (2014) have attempted to address distance-related issues at a national level using bilateral cross-country data. However, it applied a controlled geographical variation, making it difficult to justify how physical distance directly affects the development of migration intention. In rural Europe, for example, well-established spatial connectivity shortens the required travel time between places, making physical distances more negligible than those in other continents such as Asia or Africa (Cantu-Bazaldua 2021; Cattaneo et al. 2021). It could arguably affect the build-up process of migration intention as prospective destinations are not physically difficult to access from the places of origin. Moreover, urban areas are often taken as starting points for measuring remoteness (Faulkner and French 1983). This could be problematic as the build-up process of migration intentions occurs in rural areas as the places of origin. Rural areas have diverse and unique conditions as the product of their specific geographical locations, making research focusing on a single location or a controlled geographical distance unable to discover disparities in the location-specific attributes of prospective rural migrants between different places of origin. In the literature, the only study that has addressed variations in remote rural conditions limits its observation framework to socio-economic aspects (Deshingkar 2010), making it difficult to understand how variations in the physical conditions of villages affect migration. In general, geographical (physical) distances between places of rural origin and prospective destinations have not been adequately addressed as a central issue. If the geographical distance is not precisely posited at the intersection of migration intention and physical migration, it could produce biased results as physical distance might alter the perspectives of prospective rural migrants on their intentions to migrate.

Therefore, it is necessary to investigate the influence of distance-focused issues during the building up of migration intentions. Particularly, there is no solid evidence from less developed countries where migration issues still prevail in shaping national development strategies (de Haas 2006; Lerch 2020). Additionally, the investigation must consider a comparison between multiple places of origin to discover the influence of location-specific attributes of prospective rural migrants. This study, therefore, aims to investigate the effect of physical distances to prospective destinations on the intention to migrate among prospective rural migrants living in different places of origin. In this sense, it will produce a location-dependent migration intention of villagers. The underlying thought centers on whether differences in distance between origins and destinations would lead to different migration intentions. This study attempts to reveal how sensitive the building up process of migration intention is to distance as a geographical variance. In achieving the objective, this study addresses the following research questions:

- **RQ1.** Do place attachment (PA), migration intention (MI), and information sources (IS) differ across village locations in reducing obstacles towards migration?
- **RQ2.** Do differences in the remoteness of observed villages produce distinct correlational values among driving forces towards the intention to migrate?

## 2. Literature Review

### 2.1. Rural Emigration and Place Attachment

Rural emigration has long been a widely recognized phenomenon globally, making it a staple issue in the pursuit of sustainable rural development (Christiaensen et al. 2011; Tianming et al. 2018; Le et al. 2021). Although it is a classic phenomenon in rural-related discourses, the pattern of physical mobility among villagers to various destinations has continuously evolved. In general, rural-to-urban migration covers most rural out-migration patterns (Alamid and al Mamunid 2022; Marta et al. 2020). On the other hand, studies have observed a significantly increasing number of rural outmigration to foreign countries (Castles 2018; Haas et al. 2019). The imbalance of domestic demand and supply of the labor force in destination countries appears to be a prominent factor for the consistent

increases (Docquier et al. 2014). To some extent, the cross-country phenomenon pulls rural outmigration into being part of international migration. Recently, however, rural-to-rural migration has emerged as a significant trend, perhaps becoming more favorable than rural-to-urban migration (Pardede et al. 2020). In fact, it may also occur as domestic or international moves. In terms of capital spending, international emigration typically requires enormous capital to spend, including financial capital, among rural emigration typologies (Prayitno et al. 2018).

Technically, migration results from interactions among multiple driving aspects, including factors associated with origin and destination, physical/non-physical intervening obstacles, and subtle personal aspects (Lee 1966). In the literature, scholars often use economic approaches to understand human mobility (Castelli 2018; Marta et al. 2020; Prima and Khoirunurrofik 2019; Todaro 1969); however, studies on personal behavior in rural mobility are arising (Dandy et al. 2019; Pedersen 2018). Of the emerging behavioral research, studies on place attachment have recognized personal behavior as a contributing factor to rural migration along with its complexity (Barcus and Brunn 2009). Basically, a place to stay relates to the deep consciousness of an individual. The personal meaning of the place satisfies their taste and living system, which eventually influences their staying preferences. As a result, stay experiences that match individual stay preferences produce an attachment to the place. In practice, multiple factors such as the duration of stay and the multidimensional intensity of place–person interactions produce varying degrees of place attachment. Conceptually, a strengthened place attachment would reduce the likelihood of migration (Relph 1976). In terms of intergenerational trends, younger villagers tend to have a weaker place attachment than older villagers (Priatama et al. 2019). Consequently, older villagers stay considerably longer and have more intense interactions with their villages.

### 2.2. Mainstream Discourses on Migration Intention

In recent decades, research on migration intentions has emerged rapidly. Studies have considered migration intention an essential and initial clue of migration in the process towards an actual migration (Wanner 2020). Basically, migrating people have a prior intention for the migration. The degree of attachment that affects migration varies and tends to prevail for an extended period. A recent study discovered that weaker place attachment tends to yield a lower aspiration to stay (Priatama et al. 2019). However, intentions may not act as the sole factor affecting actual migration. It is also sensitive to rational elements such as the labor market or family constraints (Wanner 2020). Therefore, migration intention is not immediately translatable into actual migration (Abdelwahed et al. 2020). In addition, it is interesting to learn that migrants maintain an attachment to their places of origin even after they have migrated. Staying in destination areas for a prolonged period would never completely diminish the attachment to their places of origin (Njwambe et al. 2019). In practice, long-term and permanent migrants keep feeling attached to their physical origins (Easthope 2009).

Furthermore, the determinants of migration intention in rural areas vary. In general, there are socioeconomic, demographic, and information factors in conjunction with other personal factors (Meyer 2018; Moon et al. 2010; Perez-Barbosa and Zhang 2017; Seyfrit et al. 2010; Wolfe et al. 2020). In the literature, research on the determinants of migration intention mainly focused on economic aspects. This may relate to economic models typically used in various migration research (Todaro 1969; Žičkutė and Kumpikaitė-Valiūnienė 2015). Recently, scholars have started to pay more attention to social aspects, including place attachment. Reciprocal influences observable between a physical place and migration intention shade a new perspective in migration discourses. In practice, human–human interactions within a rural community affect place attachment (Brown and Raymond 2007; Raymond et al. 2010). Moreover, ecological interactions between the community and its surrounding environment also influence place attachment (Armstrong and Stedman 2019; Colley and Craig 2019; Raymond et al. 2010). In both interactions, the length of stay of an individual influences the affection rooted in a place, leading to an attachment (Scannell and Gifford 2010).

### 2.3. Distance and Migration Intention

Asides from the origin-focused factors, scholars have acknowledged the distance between the places of origin and destination as a determinant of migration (Bogue and Thompson 1949; Schwartz 1973). In general, a farther distance yields higher barriers toward the formation of migration intentions (Maleszyk and Kędra 2020), leading to a lower possibility for the migration process to occur (Lee 1966). In practice, any prospective migrant cannot avoid the influence of physical distance to a prospective destination on the making of one's migration decisions (Roca Paz and Uebelmesser 2021). During the decision-making process, distance acts as an external and uncontrolled factor affecting the underlying formation of migration intentions. The underlying mechanism involves internal factors such as information acquisition. Conceptually, distance dims information on a prospective destination (Lee 1966), making an information acquisition challenging to occur. Consequently, a farther distance may physically impede prospective migrants from immediately deciding on migration (Schwartz 1973). This triggers the need for advanced information delivery to help acquire information, which eventually leads to a reduced obstacle for migration (Onitsuka and Hidayat 2019).

Another internal factor influencing the underlying mechanism is place attachment. The degree of place attachment is sensitive to geographical proximity. Gieling et al. (2019) discovered an increased degree of place attachment under a farther distance to an urban area. In rural-to-urban migration, it implies a distance-affected strength of migration intention. However, it is essential to note that they conducted the research in a developed country (The Netherlands), which has considerably advanced public services. Another study suggested that public service availability contributes to the degree of place attachment (Taniguchi et al. 2012). The results may differ in less developed countries where rural areas possess inadequate public services and limited spatial connectivity (Sandee 2016). In conjunction with public service availability, spatial connectivity induces human mobility (Gustafson 2014), suggesting distinct strengths of place attachment for those living at different proximities to a destination. Under this circumstance, social channels grow organically to offset obstacles from physical barriers. Relevant to the connectivity issue, the channels help villagers reconsider their place attachment, which, after a formed intention, leads to their decision to migrate or not to migrate.

### 2.4. Migration Intention and ICT

In practice, the social channels of an individual form a social network in which the individual participates in the more extensive social networks through layers of connections. In the case of rural emigration, an organic social network typically emerges during communications between prospective migrants with rural-originated people living/ever living in a prospective destination. In fact, establishing communication contributes significantly to the formation of migration intention (Hidayati 2018). Technically, communication acts as an activity to gather information, leading to the acquisition of information about a prospective destination. In the early stages of migration, older information dissemination media (e.g., television and radio) are still essential for acquiring information (Farré and Fasani 2011). Recently, active/former migrants and prospective migrants have started to utilize a contemporary communication medium through various social media platforms (Dekker et al. 2016; Dekker and Engbersen 2014; Grubanov-Boskovic et al. 2022; McGregor and Siegel 2013). Simultaneously, older information gathering methods and platform-induced communication allow prospective migrants to form comprehensive information acquisition. In contrast, communication with prospective migrants helps active migrants maintain an attachment to their places of origin (Ozkul 2013).

In the modern information era, communication through social media platforms relies on the Internet as its underlying technology. Studies have found significant contributions of the Internet in the formation of migration intention (Moon et al. 2010; Thulin and Vilhelmson 2016; Vilhelmson and Thulin 2013). In fact, the Internet has made conventional media, including radios and newspapers, less favorable (Deloitte 2018; Ji 2019). However,

the digital divide between rural and urban areas (Hadi 2018) has made some older technologies (e.g., television and radio) prevail in shaping social activities in rural areas (Olken 2009). Governments have vigorously implemented various programs to increase Internet coverage to remote areas (Philip et al. 2017; Wilopo and Fitriati 2016). As Internet infrastructure requires significant financial and time investments (Rumata and Sastrosubroto 2021), Internet service providers prefer to build the infrastructure in urbanized areas that promise increased and faster profits (Tabor and Yoon 2015). In that sense, the Internet offers both an opportunity and challenges for rural areas, requiring prospective migrants to take full advantage of any Internet access available in their areas. Consequently, they might need to combine online and offline information sources.

### 2.5. Research Variables and Model

Scholars consider migration a reasoned action, requiring an intention to start sparking (Abdelwahed et al. 2020). Practically, intentions in a migration process go beyond migration aspiration. It is a solid sign implying that a decision is underway closer to the actual migration. Since this study focuses on the formation of migration intention as an observable critical stage in a migration decision-making process, the research model (Figure 1) sets migration intention as the dependent variable. As independent variables, this study first takes the place attachment variable. In previous sections, the literature review suggests place attachment as a critical predictor of rural mobility. Scholars have found that place attachment is established from a wide array of variables and has different mechanisms (Lewicka 2011; Raymond et al. 2010; Williams and Vaske 2003). Since this research observes the immediate influence of place attachment on migration intention, this study adopts place attachment as a singular phenomenon (Shamai and Ilatov 2005) to conclude the perceived attachment of villagers to their village.

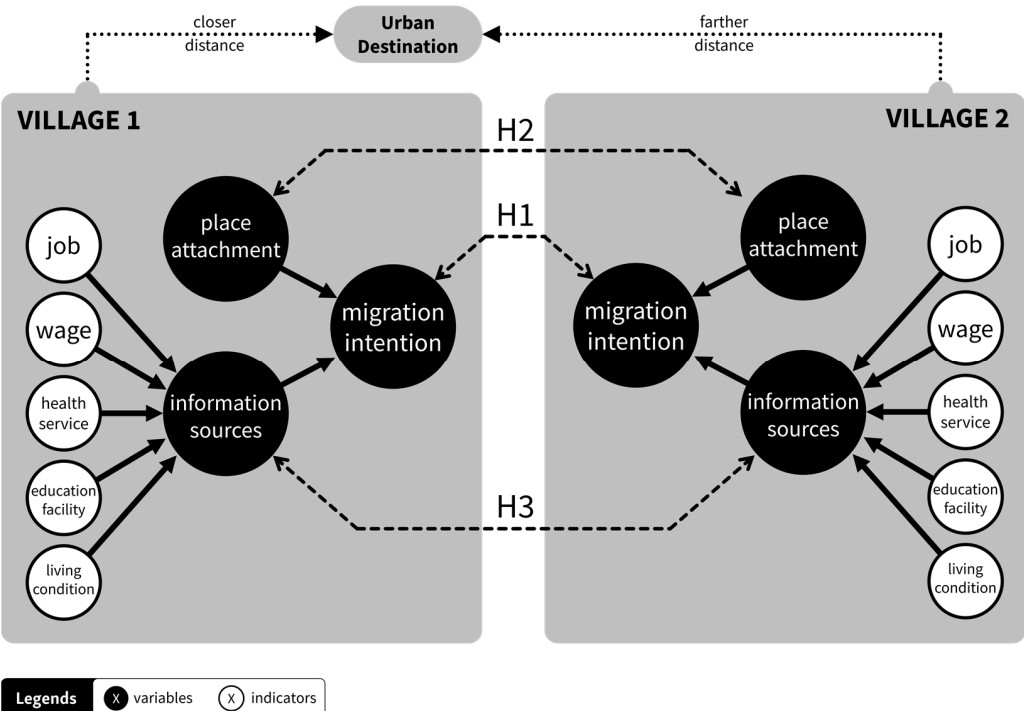

**Figure 1.** Research model and hypotheses.

On the other hand, the other independent variable observed in this study relates to information. Conceptually, information gaps are considered an intervening obstacle of migration due to the distance between an origin and a destination (Lee 1966). To some extent, it has reduced the chances for prospective migrants to learn about the place of destination. As aforementioned, prospective rural migrants utilize various ways to collect

information on prospective destinations. In this modern information era, printed media (e.g., newspapers or magazines) are less favorable (Thiel 1998). Despite being limited in number, however, they are still actively in use. In this study, information sources cover non-Internet (direct conversation, newspaper, television, radio, and magazine) and Internet-based sources (websites and social media). Regarding the types of information gathered on prospective destinations, this study focuses on information regarding jobs, wages, living conditions, health services, and education facilities (Demiralp 2009; Jones 1981). As they are part of information seeking through information sources, this study suggests indirect influences from the information types towards migration intention.

In addition, the literature review has mentioned the relationship between the build-up process of migration intention and personal factors. Thus, this research gathers the sociodemographic characteristics of villagers to represent the state of their personal conditions, which may affect their rural mobility. The sociodemographic variables in this study include, among others, gender, age, formal education level, and monthly income. As they are not decisive factors affecting the formation of intentions (Blank et al. 2018; Marta et al. 2020; Nisa et al. 2020), this research treats the sociodemographic factors as control variables that help enhance discussions over the main results.

### 2.6. Proposed Hypotheses

Based on the literature review, this research firmly considers that place attachment and information sources contribute to migration intention (Hiwatari 2016; Meyer 2018; Moon et al. 2010; Thulin and Vilhelmson 2016; Wolfe et al. 2020). The research model (Figure 1) suggests a two-side influence towards migration intention. On the one hand, place attachment emerges in the direct interactions between prospective rural migrants and their places of origin (villages). On the other hand, information emerges in the indirect interactions between the villagers and prospective destinations through various information sources. In the case of the migration intention of prospective rural migrants, this study presumes an inverse correlation between place attachment and migration intention. In short, the research model assumes that a stronger attachment implies a stronger intention to stay in the villages. Meanwhile, this study presumes a linear correlation between information and migration intention. The research model assumes that a more robust supply of information on a prospective destination implies a stronger migration intention. As information access is sensitive to the distance between the place of origin and a destination (Schwartz 1973), the physical distance between the places acts as an interfering issue in the entire migration process (Lee 1966).

**H1.** *Migration intention differs between prospective migrants in different villages.*

Considering the differences in the physical distance between a prospective destination and multiple rural areas, the research model might produce different results for different villages. However, past studies on migration intention typically focused on a single spatial level. This study compares rural areas with different characteristics to reveal a pattern of migration intention and its driving forces (Vilhelmson and Thulin 2013). The multi-case observations target the same level of governance to deliver comparable results. Therefore, this study proposes the first hypothesis (H1).

**H2.** *Place attachment differs between those living in different villages.*

As mentioned before, place attachment is an observable predictor of migration intention. Villagers with weaker place attachment tend to have stronger intentions to migrate (Priatama et al. 2019). As it measures how people attach to a particular place, geographical locations relate consequently to the degree of attachment (Berg 2020). It implies the sensitivity of place attachment to physical distances (Gieling et al. 2019). Therefore, this study proposes the second hypothesis (H2) to compare two areas located at different geographical distances from their nearest prospective destination.

**H3.** *Information (sources and types) differ between those living in different villages.*

Despite the fastest and most comprehensive information available on the Internet (Dimmick et al. 2011), inadequate Internet infrastructure and public services in rural areas (Correa and Pavez 2016) urge prospective migrants to combine online and offline information sources. As information delivery is sensitive to the distance between the place of origin and a destination (Schwartz 1973), this research assumes that the combination of information sources and types differs between villages with different distances to their nearest prospective destination. Thus, this study proposes the third hypothesis (H3).

## 3. Methodology

### 3.1. Research Design

This work follows a research design (Figure 2) that covers the entire research process. In the beginning, this study establishes an introductory narration to propose the research gaps and objectives. The literature review process acts as a follow-up to the background story for delivering the selection of observed variables (migration intention, information sources, place attachment, and sociodemographic variables), their presumed correlations in the research model (Figure 1), and the proposed hypotheses (H1–H3) for a comparative study. Next, this study selects case studies for the comparative study and the sampling process. The idea of case selection in this work centers on proposing comparable villages as the observed cases, while the sampling discovers the required sample size for each village based on a proven calculation method. After that, this study conducts data gathering in four substages, i.e., questionnaire development, distribution, and collection, followed by validation of the responses. A self-administered questionnaire containing questions on the observed variables is distributed directly to the respondents in this research stage. This stage allows the respondents to fill in the questionnaire before being collected. During the collection process, surveyors help respondents who request assistance to fill in any unanswered questions. This ensures the returns of completed responses, allowing us to validate the responses by applying face validation (Connell et al. 2018) for obtaining additional information from the respondents. Then, the last stage focuses on data analysis to process the collected data. This study employs statistical methods to test the hypotheses. First, this study applies a group comparison between two case studies using the Mann–Whitney U test, ANOVA, and the Brown–Forsythe test on each variable (i.e., place attachment, migration intention, information sources, and sociodemographic variables). In addition, this study applies the structural equation model partial least square method (SEM-PLS) to observe interactions between the observed independent and dependent variables. Then, SEM-PLS results of the cases are compared.

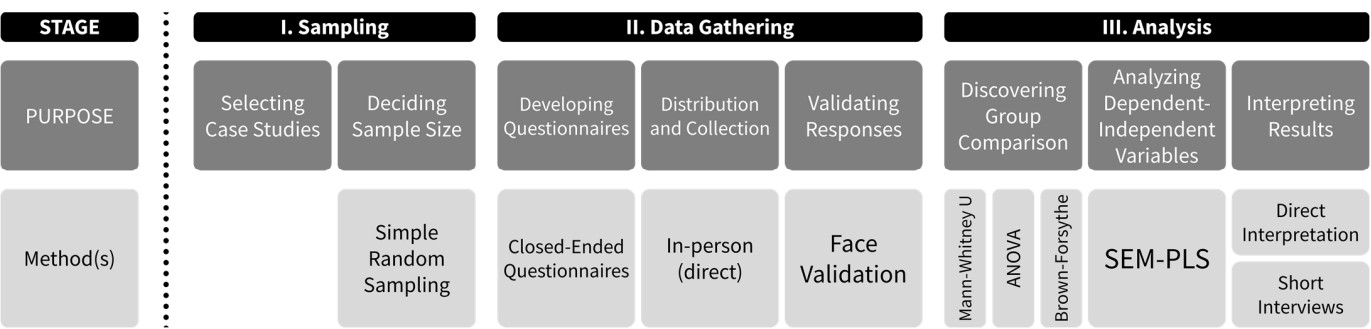

**Figure 2.** Research design.

### 3.2. Case Studies

As the world's fourth most populous country, Indonesia considers internal and international migrations a critical issue at various levels of governance (Pardede et al. 2020; World Bank 2017), including rural areas. This study aims to observe the phenomenon of rural-to-urban migration in the country. As most rural regions typically have inadequate information infrastructure and public services (Correa and Pavez 2016), this study attempts

to observe rural areas with similar characteristics on public services for comparability reasons. Moreover, this study selects rural areas that represent an extreme difference in their distances to an urban area (closest and farthest to their nearest prospective destination) to deliver the most evident differences for the observed variables. Then, this research prefers case studies with similar development conditions according to the Village Development Index (VDI) applicable in the country (Statistics Indonesia 2019). As villages nearer to the center of economic growth are more likely to have better access to infrastructure and public services (Toteng 2009), similar VDIs ensure a validated state of development for each case, and confirm an equal comparison between the cases.

Considering these reasons, this study selected two villages located in two different districts in Malang regency, East Java province, Indonesia. The villages are generally close to Malang city as their nearest prospective destination for rural-to-urban migration (Figure 3). Along with Surabaya City as the province's capital region, Malang City serves as a center of economic growth (Government of East Java Province 2011). The first case is Watugedhe village in Singosari District, located within a 10 km radius from Malang City. It is situated as a peri-urban village and is the closest village to Malang City, representing rural areas closer to the destination. The second village is Arjowilangun village in the Kalipare district. The second case is located at about 70 km distance from Malang City. It is situated as a remote village and is the farthest located village (within Malang regency) to Malang City, representing those farther from the destination. According to Statistics Indonesia (2019), the two villages have similar development indices, indicating comparable development conditions of infrastructure and public services.

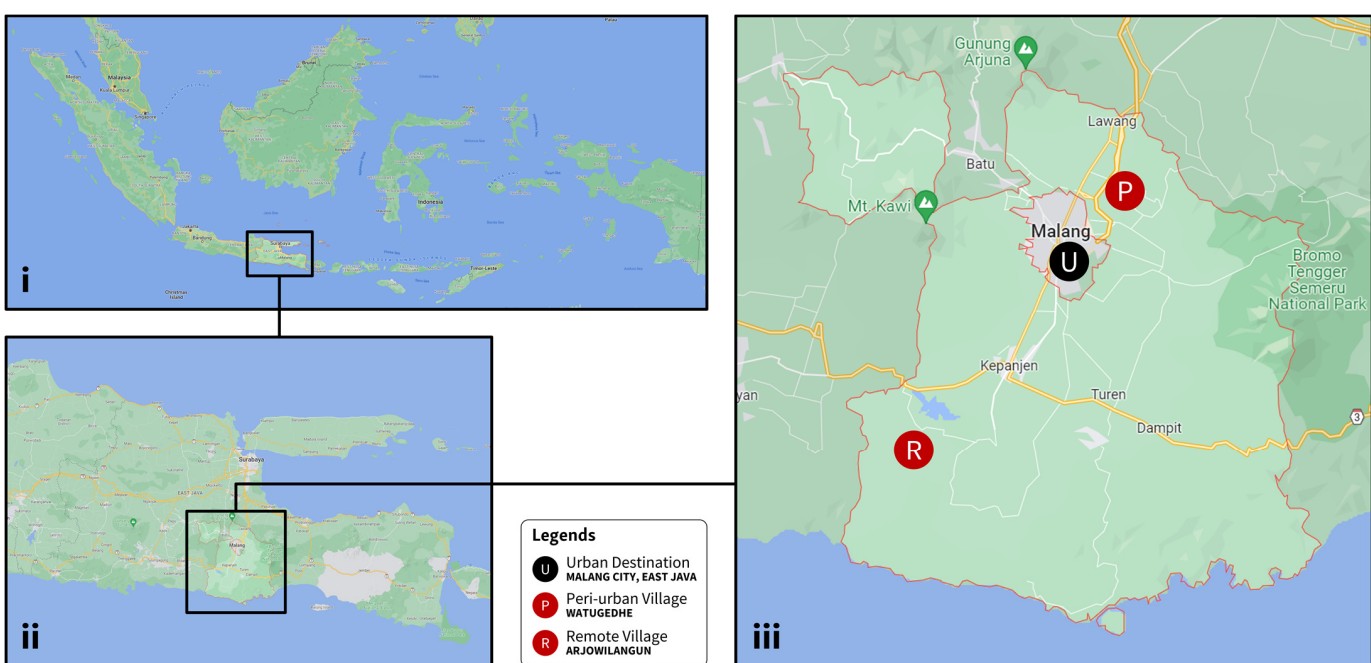

**Figure 3.** Locations of the cases (P/R) and urban destination (U) in the Malang region (**iii**), East Java province (**ii**), Indonesia (**i**).

According to Statistics Indonesia (2018, 2020a, 2020b), Watugedhe is home to 10,469 villagers. About 46% of the rural dwellers work at local factories. In terms of landscape, the village is generally located on plain lands. Based on the spatial plan, this 138-ha village is designated as an industrial area, which stands side-by-side with maintained agricultural lands and activities. However, agricultural land is the most prominent land use, which is followed by industrial land use and settlements. Despite the vast size of agricultural land, farming is not the village's primary occupation. Only 2.56% of the total villagers are working as farmers. Meanwhile, Arjowilangun village is basically an agricultural area and

home to 16,498 villagers. Most villagers in Arjowilangun work as farmers, covering 52.69% of the village's total labor force. In 2020, approximately 1003 ha of total land in the village was used for agricultural purposes, including wetland and plantation. The lands span from plain to hilly and mountainous areas, resulting in the diverse topographical contours of the village. As an agricultural village, Arjowilangun produces timber and sugarcane in addition to rice as its primary agricultural product.

### 3.3. Sampling and Data Collection

This study applies a validated sampling method to ensure that the sample size and selected respondents are statistically adequate, reflect the observed population, and fit to the research objective. This research uses a simple random sampling to determine the minimum sample size in every village. It follows the equation formulated in the work of Krejcie and Morgan (1970), which suggests a 95-degree accuracy (Equation (1)).

Among villagers, age corresponds to the length of stay, by which younger residents show a lower place attachment than older villagers (Dallago et al. 2009). By also considering an intergenerational gap between younger and older generations in the use of the Internet, the unit analysis of this study is the younger generation as a subset of the village's entire population. Therefore, the sampling calculates and chose respondents from young villagers in each village. In this study, villagers aged 15–24 are eligible to be the respondents. According to Statistics Indonesia (2018, 2020a, 2020b), the eligible population in Watugedhe and Arjowilangun are 885 and 1874 young people, respectively. This study applies Equation (1) to get the minimum sample size for each village, resulting in 300 respondents from Watugedhe village and 330 respondents from Arjowilangun village. During data collection, selected respondents are requested to complete close-ended questionnaires. They are also allowed to ask for assistance during the process. The self-administered questionnaires are distributed to the respondents with a 100% return rate. A one-week period is provided for respondents to fill in the survey questions. During the collection of survey responses, this study conducts short interviews to ensure respondents' understanding of the questions, validate their answers to the surveys, and discover additional valuable information to enrich the discussion.

$$s = X^2 NP(1 - P) \div d^2(N - 1) + X^2 P(1 - P) \tag{1}$$

where

$s$ = required sample size
$X^2$ = chi-square for 1 degree of freedom at the desired confidence level (3.841)
$N$ = population size
$P$ = population proportion (assumed at 0.5 for maximum sample size)
$d$ = degree of accuracy expressed as proportion (0.05).

### 3.4. Research Instruments

Table 1 shows the data targeted in this study. All data in the questionnaire are ordinal. This study measures migration intention using a five-level Likert scale, allowing for an observation on the degree of migration intention. Score 1 (minimum) represents no intention, while Score 5 (maximum) represents a "very strong" (firm) migration intention. Those with a fragile migration intention can choose the middle score (3). A fragile intention refers to the state at which a respondent may gain or lose one's migration intention quickly at any time. It provides a middle ground between migration-leaning intentions (Scores 4–5) and stay-leaning intentions (Scores 1–2). Furthermore, Berg (2020) stated that there is a degree of place attachment in response to the geographical location of a place of origin. Instead of using multiple questions to measure the degree, this research adopts a general place attachment question (Shamai and Ilatov 2005) to conclude the degree of place attachment for each respondent. It delivers a simplified survey to the respondent, especially for the place attachment part, ensuring complete responses.

**Table 1.** Data and analyses for the proposed hypotheses.

| Hypotheses | Data | | | | Analysis Tool |
|---|---|---|---|---|---|
| | **Migration Intention** | **Place Attachment** | **Information Sources** | | |
| | | | **Internet** | **Non-Internet** | |
| H1 | 1 → not at all<br>2 → no intention<br>3 → have a low-level migration intention and possibly lost it anytime with a weak reason<br>4 → strong intention<br>5 → very strong intention to migrate | - | - | - | Mann–Whitney U-test |
| H2 | - | 1 → have no attachment to the village at all<br>5 → have a very strong attachment to the village | - | - | Mann–Whitney U-test |
| H3 | - | - | • Two options (website and social media)<br>• Sum of used information sources in every type of information<br>• (0 = not using Internet to collect information; 2 = using two information sources) | • Five indicators (face-to-face conversation, TV, radio, magazine, newspaper, and other non-Internet media)<br>• Sum of used information sources in every type of information<br>• (0 = not using non-Internet media to collect information; 6 = using 6 information sources) | ANOVABrown-Forsythe test |

In addition, this study measures information sources through five indicators related to the prospective destination (jobs, wages, living conditions, health services, and educational facilities). In terms of information sources, this study considers face-to-face conversations and non-Internet-based communications (e.g., voice calls or text messaging) as actual conversations. As social media platforms are evolving, this study treats messaging applications (e.g., WhatsApp or Line) as a form of social media (Alsanie 2015; Dahdal 2020). Moreover, this study provides the option "others" if a respondent uses information sources not provided in the questionnaire. However, this research intentionally excludes traditional mail (via post), considering its limited use in the villages. The minimal use is because telephone connection is available in the village. Then, this research distinguishes between Internet-based and non-Internet information sources. Therefore, this study sums up the sources in each indicator. For instance, a respondent may collect job information from Internet-based and non-Internet sources. If the respondent uses two non-Internet sources and one Internet source, the respondent should answer "three" (3).

*3.5. Data Analysis*

Instead of observing the model for both villages, this study compares the states of each variable between two observed villages. Therefore, the model fit indices are less relevant (Kock 2019). In practice, the analysis first investigates differences in migration intention (dependent variable) between the two villages (H1). Then, the analysis compares the villages regarding the contributions of place attachment (independent variable; H2) and information sources (independent variable; H3) towards migration intention. In detail, the analysis begins with applying descriptive statistics to lay the foundation of understanding the data prior to hypothesis testing. Furthermore, this research conducts data pre-processing to fit the data requirements for the statistical analyses. Prior to conducting a group comparison on each variable between the two cases, this research classifies the scores for migration intention and place attachment based on the sociodemographic characteristics of the respondents. Then, this study applies the Mann–Whitney U test for the first and second hypothesis testing (H1 and H2) as they are single-indicator tests. For the multi-indicator third hypothesis (H3), this study applies the analysis of variance (ANOVA) for indicators meeting ANOVA assumptions. For indicators with heterogeneity (violates ANOVA assumptions), the analysis tests the third hypothesis by applying the Brown–Forsythe test (Tomarken and Serlin 1986). Then, this research uses the WarpPLS software to conduct an SEM-PLS analysis on each village for comparing the SEM-PLS models of the villages.

## 4. Results

*4.1. Demographic Characteristics of Respondents*

The surveys involve 630 randomly selected young villagers (15–24 years old) from the two observed villages. Of the respondents, 300 respondents (out of 885 young villagers) are from Watugedhe (peri-urban village). The rest (330 respondents) are from Arjowilangun village (out of 1874 young villagers). Among those living in Arjowilangun (remote village), six respondents appear to be non-active Internet users. Table 2 shows the sociodemographic profiles of all respondents. It appears that each case is unique. The villages have different compositions of gender, age group, and education. However, data on occupation and income distribution show similar patterns. Looking at the gender composition, most of the respondents in the peri-urban village identify as males. On the contrary, most of the respondents in the remote village identify themselves as women. To better understand the differences among age cohorts, this study follows Steinberg (2020) to classify respondents into three age groups: middle adolescence (15–18 years old), late adolescence (19–22 years old), and young adults (23–24 years old). The distribution for the peri-urban village is primarily middle and late adolescence. In contrast, most of the respondents from the remote village are in the middle adolescence groups, with dramatic decreases in number for the late adolescence and young adult groups. In terms of occupation, most respondents

in both villages are students (66.6% and 67.0% for the peri-urban village and the remote village, respectively). Interestingly, several school-age respondents stop going to school to work in various occupations. As a result, the composition of age groups does not reflect the distribution of occupations. For consistency purposes, this research treats students' pocket money (monthly average) as income. The average income of respondents for the peri-urban village is almost double that of the remote village. However, the average income in both villages is much lower than the regency's minimum wage (Government of East Java Province 2020). Moreover, the non-student income (wages) in both villages is much higher than the income of students (pocket money). The ratios are almost quadruple for the peri-urban village and triple for the remote village. The average income of students in the peri-urban village (IDR 550,850 → ±USD 38.03) is higher than for the students living in the remote village (IDR 355,599.08 → ±USD 24.54). College-age respondents contribute more to the average income because their income is considerably higher than that of the students.

**Table 2.** Sociodemographic characteristics of the respondents.

| Characteristics | Category | Peri-Urban Village | | Remote Village | |
|---|---|---|---|---|---|
| | | *n* | % | *n* | % |
| Gender | Male | 142 | 47.3 | 174 | 53.7 |
| | Female | 158 | 52.7 | 150 | 46.3 |
| Age | 15–18 | 124 | 41.3 | 208 | 64.2 |
| | 19–22 | 127 | 42.3 | 80 | 24.7 |
| | 23–24 | 49 | 16.3 | 36 | 11.1 |
| Education | No education | 0 | 0.0 | 1 | 0.3 |
| | Elementary school | 1 | 0.3 | 25 | 7.7 |
| | Junior High School | 137 | 45.7 | 190 | 58.6 |
| | Senior High School | 149 | 49.7 | 98 | 30.2 |
| | College | 13 | 4.3 | 10 | 3.1 |
| Occupation | Jobless | 8 | 2.7 | 25 | 7.7 |
| | Junior High School Students | 18 | 6.0 | 36 | 11.1 |
| | Senior High School Students | 118 | 39.3 | 165 | 50.9 |
| | College students | 64 | 21.3 | 16 | 4.9 |
| | Housewife | 1 | 0.3 | 13 | 4.0 |
| | Freelancer | 7 | 2.3 | 16 | 4.9 |
| | Farmer | 2 | 0.7 | 3 | 0.9 |
| | Private Company Employer | 20 | 6.7 | 20 | 6.2 |
| | Government Employer | 19 | 6.3 | 11 | 3.4 |
| | Entrepreneur | 43 | 14.3 | 19 | 5.9 |
| Income [IDR] | <250,000 | 37 | 12.3 | 121 | 37.3 |
| | 250,001–500,000 | 99 | 33.0 | 123 | 38.0 |
| | 500,001–750,000 | 1 | 0.3 | 38 | 11.7 |
| | 750,001–1,000,000 | 100 | 33.3 | 17 | 5.2 |
| | 1,000,001–1,250,000 | 0 | 0.0 | 3 | 0.9 |
| | 1,250,001–1,500,000 | 33 | 11.0 | 5 | 1.5 |
| | >1,500,000 | 30 | 10.0 | 17 | 5.2 |
| **Mean *** | | 1,000,900 (±USD 69.1) | | 503,132.72 (±USD 34.97) | |

Note: * USD 1 = IDR 14,485.55 as of 26 April 2021.

### 4.2. Migration Intention

Table 3 shows the patterns of correlation between place attachment/migration intention for the two observed villages across the sociodemographic properties of the respondents. The results center on the identification of comparative differences (U-values) between peri-urban and remote villages to separately test the first (migration intention; H1) and second (place attachment; H2) hypotheses. Statistically speaking, the migration

intention and place attachment for peri-urban and remote villages significantly differ. Insignificant results appear due to inadequate data sizes for the comparisons, for example, in the comparative U-values between place attachment (H2) and those who attend/attended elementary school, college level, and those who have never attained any educational level. In general, the place attachment data for the remote village have statistically higher mean ranks than those of the peri-urban village. The comparative U-values confirm the significant differences. The results imply that respondents living in the remote village have significantly stronger place attachment than those living in the peri-urban village. In terms of migration intention, comparative U-values for the migration intention between the villages indicate a similar tendency. Looking at the mean ranks, respondents living in the remote village have a stronger migration intention compared to those living in the peri-urban village. Still, the differences are significant.

**Table 3.** Comparative U-values across sociodemographic groups.

|  | | Variable | Peri-Urban Village (PUV) | | | Remote Village (REV) | | | U-Value |
|---|---|---|---|---|---|---|---|---|---|
|  | | | Samples (*n* = 300) | Mean | S.D. * | Samples (*n* = 324) | Mean | S.D. * | |
| Migration Intention(H1) | Gender | Male | 142 | 3.06 | 0.665 | 173 | 3.51 | 1.223 | 9551.5 ** |
|  | | Female | 158 | 2.99 | 0.672 | 150 | 3.73 | 1.241 | 7587.0 ** |
|  | Age group | Mid adolescence | 124 | 3.04 | 0.759 | 208 | 3.45 | 1.084 | 10,442.5 ** |
|  | | Late adolescence | 127 | 2.99 | 0.624 | 80 | 3.68 | 1.456 | 3366.5 ** |
|  | | Young adult | 49 | 3.08 | 0.534 | 36 | 4.44 | 1.182 | 266.0 ** |
|  | Education | Never attended school | - | - | - | 1 | 2.00 | - | - |
|  | | Elementary school | 1 | 3.00 | - | 25 | 4.00 | 1.190 | 6.5 |
|  | | Junior high school | 137 | 3.04 | 0.736 | 190 | 3.46 | 1.062 | 10,298.0 ** |
|  | | Senior high school | 149 | 3.01 | 0.604 | 97 | 3.90 | 1.425 | 4227.5 ** |
|  | | Higher education | 13 | 3.00 | 0.707 | 10 | 3.00 | 1.700 | 65.0 |
|  | Intensity | Have no intention at all | 3 | 3.03 | 0.669 | 16 | 3.61 | 1.233 | 34,572.0 ** |
|  | | Have no intention | 49 | | | 42 | | | |
|  | | Somewhat have intention | 190 | | | 110 | | | |
|  | | Have strong intention | 53 | | | 39 | | | |
|  | | Have a very strong intention | 5 | | | 117 | | | |
| Place Attachment(H2) | Gender | Male | 142 | 2.99 | 0.625 | 174 | 4.01 | 0.797 | 4072.5 ** |
|  | | Female | 158 | 2.91 | 0.626 | 150 | 4.07 | 0.791 | 3358.0 ** |
|  | Age group | Mid adolescence | 124 | 2.84 | 0.603 | 208 | 4.06 | 0.768 | 3206.0 ** |
|  | | Late adolescence | 127 | 3.02 | 0.630 | 80 | 3.96 | 0.849 | 1921.5 ** |
|  | | Young adult | 49 | 3.04 | 0.644 | 36 | 4.06 | 0.826 | 336.5 ** |
|  | Education | Never attend school | - | - | - | 1 | 5.00 | - | - |
|  | | Elementary school | 1 | 3.00 | - | 26 | 3.69 | 1.011 | 5.5 |
|  | | Junior high school | 137 | 3.15 | 0.593 | 194 | 1.91 | 0.753 | 2868.5 ** |
|  | | Senior high school | 149 | 3.03 | 0.581 | 99 | 1.95 | 0.787 | 2351.5 ** |
|  | | Higher education | 13 | 2.31 | 0.947 | 10 | 2.60 | 0.843 | 50.5 |
|  | Intensity | Very low | 0 | 2.95 | 0.626 | 2 | 4.04 | 0.794 | 15,911.0 ** |
|  | | low | 58 | | | 7 | | | |
|  | | Moderate | 209 | | | 63 | | | |
|  | | Strong | 24 | | | 157 | | | |
|  | | Very strong attached | 9 | | | 95 | | | |

Note: * S.D. = standard deviation. ** significant at 0.05 level.

### 4.3. Information (Sources, Uses, and Types)

Information sources, especially those that supply information on prospective destinations, are essential for prospective migrants. Due to the distance between the place of origin and prospective destinations, accessible information sources help increase the understanding of prospective migrants over the destinations. The sources allow prospective migrants to search for important information on jobs, wages, health services, education facilities, and living conditions in the destinations. Table 4 displays the results of this study for the information source variables and indicators. Looking at the results, this study reveals that the respondents have already been utilizing the Internet to collect information on prospective destinations. Apparently, there are distinguishable patterns between the villages. Internet users in the peri-urban village have fully utilized the Internet to collect information regarding prospective destinations. However, Internet users in the remote village have not adequately utilized the Internet to get information as such, despite indicating a high Internet usage. On the other hand, this study, as before mentioned, designs the questionnaire to uncover other information sources used by the respondents (Internet users). The result indicates that only a few respondents living in the remote village utilize non-Internet information sources. The non-Internet information sources ("others") appear to include leaflets and advertising banners. Those "other" sources mainly contain opportunities to work as international migrant workers, but less information on domestic job opportunities.

**Table 4.** Comparison of the types and sources of information between peri-urban and remote villages.

| Information (H3) | | Peri-Urban Village (PUV) | | | | | Remote Village (REV) | | | | | |
|---|---|---|---|---|---|---|---|---|---|---|---|---|
| | | Yes * | | No | | S.D. ** | Yes * | | No | | S.D. ** | U-Value |
| Source | Type | n | % | n | % | | n | % | n | % | | |
| | Job | 300 | 100.0 | 0 | 0.0 | 0.390 | 241 | 74.4 | 83 | 25.6 | 0.672 | 31.642 *** |
| | Wage | 300 | 100.0 | 0 | 0.0 | 0.398 | 237 | 73.1 | 87 | 26.9 | 0.650 | 49.652 *** |
| Internet | Health service | 300 | 100.0 | 0 | 0.0 | 0.401 | 66 | 20.4 | 258 | 79.6 | 0.403 | 956.408 *** |
| | Education | 300 | 100.0 | 0 | 0.0 | 0.401 | 235 | 72.5 | 89 | 27.5 | 0.600 | 80.276 *** |
| | Living condition | 300 | 100.0 | 0 | 0.0 | 0.329 | 229 | 70.7 | 95 | 29.3 | 0.600 | 67.306 *** |
| | Job | 284 | 94.7 | 16 | 5.3 | 0.589 | 213 | 65.7 | 111 | 34.3 | 0.859 | 131.121 *** |
| | Wage | 270 | 90.0 | 30 | 10.0 | 0.670 | 161 | 49.7 | 163 | 50.3 | 0.862 | 151.798 *** |
| Non-Internet | Health service | 273 | 91.0 | 27 | 9.0 | 0.656 | 62 | 19.1 | 262 | 80.9 | 0.451 | 773.482 *** |
| | Education | 267 | 89.0 | 33 | 11.0 | 0.687 | 180 | 55.5 | 144 | 44.4 | 0.883 | 153.184 *** |
| | Living condition | 280 | 93.3 | 20 | 6.7 | 0.616 | 177 | 54.6 | 147 | 45.4 | 0.913 | 166.127 *** |

Note: * number and percentage of respondents who utilize Internet and non-Internet information source. ** S.D. = standard deviation for information sources ($0 \rightarrow$ not using; 1 to $5 \rightarrow$ sum of information sources). *** significant at the 0.05 level.

There are observable differences in the pattern of Internet use between respondents living in the peri-urban and remote villages. Respondents from the peri-urban village have utilized the Internet to gather desired information. In contrast, some respondents from the remote village have not adequately utilized the Internet for information gathering. However, respondents from both villages have used various non-Internet sources of information. In detail, the number of respondents who have not used non-Internet information sources is higher in the remote village than the peri-urban counterpart. The trend appears significant for any type of information based on the mean ranks. Furthermore, the analysis of variance (ANOVA) applies for the information on health services (from Internet sources) and education facilities (from non-Internet sources) as they meet ANOVA assumptions. Meanwhile, the rest of the indicators show heterogeneity, thus violating ANOVA assumptions. For those indicators, the analysis applies the Brown–Forsythe test. In general, the results of these analyses show significant differences between the villages for all sources and types of information. This implies that the use of information sources differs between the peri-urban and the remote villages. Moreover, the results indicate that

respondents living in the remote village utilize more information sources than those living in the peri-urban village. The trend occurs for all information types except for information on health services using non-Internet sources.

### 4.4. Correlation between Migration Intention, Place Attachment, and Information Sources

Instead of an in-depth investigation of the model, this study focuses on the direct interactions between independent and dependent variables under observation. Figure 4a,b show the comparison of the two cases. The location-specific models indicate 7% and 10% of variances in migration intention for the peri-urban and remote villages, respectively. In the analysis, this study sets all variables as reflective ones. Looking at the results, they pass the reliability (>0.7) and validity measurements (>0.5) with Cronbach's alpha scores ranging from 0.705 to 1.000, and the scores of average variances extracted ranging from 0.608 to 1.000. As migration intention and place attachment are single-indicator variables, their reliability and validity measurements reach as high as 1.000. Between the two cases, information sources appear to negatively impact migration intention variable significantly. Meanwhile, place attachment insignificantly affects the migration intention of respondents living in the peri-urban village, but oppositely affects the intentions of those living in the remote village.

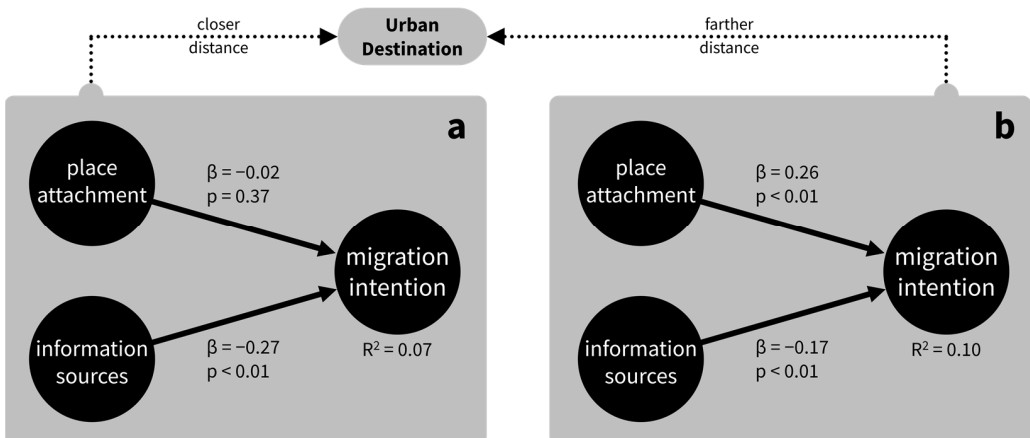

**Figure 4.** Correlation models for (**a**) peri-urban village and (**b**) remote village.

## 5. Discussion

### 5.1. Migration Intention (Dependent Variable)

Distance is considered an essential factor throughout the process of migration. Aside from becoming a physical barrier during actual migration, distance affects information flow, which is essential during processes preceding the actual migration. It implies the entire migration as a long and complex process involving multiple stages, from the spark of intention to the settling in after physically moving to the destination (Czaika et al. 2021). Considering the complexity of human mobility, interregional migration has become a prominent issue as it involves considerably distant settings between an origin and a destination within the same jurisdiction. In the case of rural-originated migration, rural-to-rural migration has gradually shown more significant growth than rural-to-urban migration (Pardede et al. 2020). However, migration from rural to urban areas is becoming critical for the sustainable development of the distinctly established yet reciprocally dependent regions (Tianming et al. 2018). Unfortunately, the trends indicate that urban areas gain significant populations from rural migrants, making rural areas lose critical human forces to sustain rural development. By 2045, 70% of the world's total population is projected to reside in urban areas (Roberts et al. 2019).

This research proves the high intention to migrate among village residents. In this sense, all rural dwellers are prospective migrants, making it essential to further investigate into the formation process of their intentions. It appears that high migration intentions

apply to villagers living in either peri-urban or remote villages. Interestingly, this research discovers higher migration intentions of villagers living farther from an urban area. Despite living closer to the comparably nearest prospective destination, villagers living in a peri-urban village have a weaker intention to migrate. In general, it appears that villagers from a remote village have a stronger migration intention as they consider multiple reasons for their migration. Whether it is to improve their livelihoods, access better public services, or to obtain better economic conditions, emigrating to urban areas is consequently a reasonable decision. The finding strengthens concerns in prior studies on the adverse impacts of rural emigration on rural origins (Kanbur and Zhuang 2013; Putra et al. 2020). The concerns include, but are not limited to, decreased labor forces (Akram et al. 2017; Li 2015; Williams and Paudel 2020), disturbed agricultural activities (de Brauw 2019), abandonments of rural land (Gray and Bilsborrow 2014; Xu et al. 2019), and widened rural–urban inequality (He 2012; Kundu and Pandey 2020).

Besides pull factors from urban destinations, push factors such as limited public services and infrastructure in rural areas, have driven villagers to move out from their rural origins (Brueckner and Lall 2015; Dustmann and Okatenko 2014). This study confirms that by looking at the low village development indices in the villages. The indices depict a general overview of similarly under-developed conditions of public services and infrastructure in peri-urban and remote villages (Statistics Indonesia 2019). Aside from the secondary information, the statistical results indicate that prospective rural migrants living farther from prospective urban destinations experience a slightly lower development index. The farther distance from urban areas seems to affect the under-performing development, making villagers living in remote villages have stronger migration intentions than those living closer to urban areas. It confirms typical city-oriented rural development styles adopted by less developed countries (Lin et al. 2019; Zhang and Lu 2018). Consequently, villagers living closer to urban areas have less urgent needs for rural-to-urban migration as they can access necessary facilities/services anytime with fewer resources. For those living farther from urban areas, the need to move to locations with better facilities/services gradually forms their migration intention. The phenomenon strengthens the positioning of distance to public services as a critical requirement for ensuring excellent services to village residents.

In addition, rural areas typically lack the economic resources and knowledge necessary to develop better rural public services and other necessary facilities (e.g., health and education). This induces a strengthening loop between the less availability of public services/facilities in rural areas, lower incentives for rural dwellers to stay in their villages, urbanizations, lesser economic activities in the villages, and lower growths to gather necessary economic resources for rural public services/facilities. In the observed country, the government has set broader service coverage of urban public services to serve rural dwellers living nearby (National Standardization Agency 2004). In general, the regulation requires urban public services and facilities to accept more villagers. Initially, the purpose was to solve the unavailability of rural public services/facilities. As accessing urban public services is inevitable due to limited services/facilities in rural areas (Wajdi et al. 2017), the regulation eventually promotes more rural-to-urban migrations for those living far from urban areas. Conversely, it gives more reasons for rural dwellers living closer to urban destinations to stay since there is no urgent necessity to emigrate from their villages. Peri-urban villages can access desired urban public services without permanent moves to urban areas located nearby.

Furthermore, the ever-strengthening cycle between rural development and emigration has made prospective migrants favor domestic interregional migration more than international migration. Since domestic migration require fewer resources than cross-country migration, the distance-induced discrepancies in rural development and resource availability allow prospective rural migrants to foster their intentions to migrate domestically (Prayitno et al. 2018). Studies on domestic migration have covered various determining factors and trends (Pardede et al. 2020) at micro or macro levels for different spatial entities.

This research further expands the body of knowledge of domestic interregional migration by discovering the distance-dependent migration intention, especially in less-developed regional/national development settings. In fact, distance-dependent rural conditions have fostered the formation of stronger distance-dependent migration intentions among young villagers. This research confirms previous studies that found younger villagers are more likely to have less intention to stay than older rural dwellers (Meitasari 2017; Priatama et al. 2019). Compounded issues in rural areas that are coupled with the more developed attractiveness of urban areas have maintained the strengthening loop of migration intentions among rural youth. Unfortunately, this strengthens concerns on the future of rurality, in which young generations may not likely be the fundamental force or the "brainware" behind future rural sustainability.

*5.2. Place Attachment and Information Sources (Independent Variables)*

Furthermore, the results of this study imply that the two observed villages have distinct characteristics. In conjunction with the explanations on migration intention, peri-urban villages have more urban characteristics than their remote counterparts. Regarding the observed independent variables (place attachment and information sources), the migration intention of prospective rural migrants living in peri-urban villages is only affected by information sources. For those living in remote locations, place attachment and information sources deliver significant impacts on migration intention in parallel. These distinct characteristics provide evidence that place attachment is not always the strongest predictor of rural emigration. This study thus extends the findings of prior literature (Gieling et al. 2019; Pedersen 2018). Notably, young villagers with lower place attachment actively use the Internet to consider their aspirations to migrate. Access to the Internet allows them to have Internet-based information sources and communication tools, adding to existing non-Internet media such as magazines and radio. Instead of entirely replacing prior technologies, this research discovers that young villagers use the Internet as a complementary technology to existing information dissemination media. It precisely confirms Dimmick et al. (2011), who suggested the incremental signs of progress offered by the Internet on top of existing technologies for information seeking.

Basically, this study does not nullify the findings of prior literature (Gieling et al. 2019; Pedersen 2018) that suggested the impact of place attachment on migration intention. However, the U-values for place attachment (Table 3) and the SEM-PLS models (Figure 4) reveal that a strong place attachment relates to a strong migration intention. The results interestingly oppose a common understanding (Petrovic et al. 2017; Priatama et al. 2019), in which weaker place attachment (to the place of origin) leads to the formation of intentions to migrate (to another place outside the rural origin). Still, the SEM-PLS models show low $R^2$ scores (peri-urban village: 0.07; remote village: 0.10), indicating that only 7% (peri-urban village) and 10% (remote village) of the total variances of the dependent variable (migration intention) are explainable by the independent variables (place attachment and information sources). It is understandable as economic motives typically drive rural emigration. This study confirms the study by Lyu et al. (2019), in which the migration intention of rural migrants is fostered by their needs to sustain their livelihoods. Remittances from active rural migrants are beneficial for relatives who still live in their rural origins (Semela and Cochrane 2019). In that sense, the translation of a strong place attachment into a firm migration intention reflects how prospective rural migrants are more attracted by pull factors (better livelihoods) from urban destinations than the attachment to their rural origins.

In addition, this study shows that the place attachment of young villagers living in remote places is significantly higher than that of those living in peri-urban locations, confirming the findings of Buchecker and Frick (2020), and Gieling et al. (2019). As their findings were based on developed countries, this study complements them by providing findings in less developed settings. Thus, both developed and developing countries show similar patterns of place attachment among villagers. Moreover, this study also discovers

the distance-dependent place attachment of young villagers. Villagers living farther from a prospective urban destination tend to have stronger place attachment than those living closer to urban areas. By explaining the geographical variations of place attachment in less-developed settings, this work complements the findings of Gieling et al. (2019) discovered in more developed regions. This study suggests that villagers living in remote areas have limited movements outside their remote villages, resulting in intensive interactions with their rural places. It confirms studies that suggested strengthened preferences to stay due to more active person–place interactions (Barcus and Brunn 2009; Pedersen 2018). In contrast, the place attachment of peri-urban villagers insignificantly contributes to their migration intention. Basically, they have better spatial connectivity (Yamauchi et al. 2011) and public service spillovers from nearby urban areas (Marshall et al. 2009). As a result, they form a multi-place attachment beyond their rural boundaries (Gustafson 2014).

On the other hand, the rapid advances of ICTs have delivered various impacts, including social and economic benefits, for rural communities (Priatama et al. 2019; Rini and Rahadiantino 2020). The phenomenon induces the Internet as an integral part of society. However, this study discovers that geographical variations also deliver challenges regarding the digital divide, confirming an existing study (Jurriens and Tapsell 2017). There is limited availability of information sources in rural areas due to the diverse geographical conditions. This has resulted in a low penetration of the Internet and a weak flow of information through the Internet. For instance, some people in remote rural areas use satellite receivers to watch television, which, as Shobaruddin (2019) stated, makes it difficult for villagers to combat poor information literacy. In parallel to the distance-dependent migration intention, this study finds that information flows take more critical roles in forming the intention. It agrees with Lee (1966), who suggested the essential position of information to address the problem of physical distance. Practically, moving from place to place requires information regarding prospective destinations. The dramatic improvements of ICTs make distances in communication and information seeking irrelevant. Instead of depending on physical visits to a prospective destination, villagers can foster their intention by using the ICTs for information gathering.

Parallel to prior studies (Moon et al. 2010; Thulin and Vilhelmson 2016), this study suggests that the use of multiple information sources and types affects the build-up process of migration intention. Prospective rural migrants living in peri-urban and remote villages have statistically distinct choices of information sources. Villagers living in peri-urban villages use more diverse information sources (both Internet and non-Internet) than those living in more remote places. The significant differences in information sources imply that peri-urban villagers have better information literacy than their more remote counterparts. In consideration of the strong dependency of information dissemination on the ICT infrastructure, this study also suggests that villages in the proximity of urban areas have a better information infrastructure than remote villages. The infrastructure issue defines the availability of information sources. Supported by an existing study (Marshall et al. 2009), peri-urban villages take full advantage of the spilled-over services and facilities from nearby urban areas. In fact, the finding emphasizes the distance-dependent trends for the choice of information sources. Interestingly, this research reveals that information sources impede migration intention. It implies that the obtained information unmasked actual conditions in prospective destinations. Gradually, prospective migrants may realize that situations in prospective destinations may not fit their expectations. It eventually impedes the formation of migration intention for prospective rural migrants.

## 6. Conclusions

Scholars have begun to pay attention to migration intention as an essential part of discourses on rural migration. In general, prior studies focused on a single spatial entity to observe the intention to migrate among prospective rural migrants. This research provides multi-case evidence on distance-dependent migration intention to fill the theoretical gap. Aside from the spatial issues, the findings of previous research leaned to trends in developed

countries, fragmenting the applicability of their findings to rural areas in less developed economies. Therefore, this work provides complementing evidence by observing the phenomenon in less-developed settings in which public services and spatial connectivity are sensitive to geographical conditions. Then, this study goes beyond prior approaches by correlating the distance-dependent migration intention to its driving forces. As a general result, migration intention is predicted not only by the place attachment of prospective rural migrants, but also by information sources/types they use/choose.

In detail, this study reveals that place attachment and information sources only affect migration intention in remote rural areas where migration intention tends to be stronger than in peri-rural areas. It suggests that place attachment is not always the primary contributor in the formation of migration intention. Place attachment in peri-urban areas is apparently lower than in remote rural areas. Meanwhile, those living in remote areas tend to have stronger place attachment and firm migration intention. The two situations differ from a common understanding that suggests weaker place attachment results in stronger migration intention. On the other hand, information sources interestingly deliver adverse impacts towards migration intention. This implies that information (sources and types) make prospective rural migrants realize the truth regarding their prospective urban destinations, impeding the formation of an intention to migrate.

If the government intends to prevent rural emigration, this study, therefore, suggests that the government focus on improving information literacy. Instead of strengthening the place attachment of villagers to their villages, introducing Internet-based and non-Internet information sources allow prospective rural migrants to reconsider their aspiration to migrate. In fact, it helps the government to control rural emigration while also doing their obligation to enhance human development for villagers. Improving information literacy requires the government to build better infrastructure and public services, making them able to strengthen rural place attachment along the way. The development may also provide economic growth to rural areas, incentivizing rural dwellers to have better livelihoods while staying in their rural origins.

This work acknowledges some limitations that also generate insights for future research works. First, this study measures place attachment using a single simple question, which cannot produce deeper insights on the formation of place attachment itself. As this study establishes the research boundary by locating place attachment as an independent variable of migration intention, we suggest that future studies focus on in-depth investigations on the formation of place attachment (place attachment as a dependent variable). Second, both case studies are in Java Island, where public services and spatial connectivity are much more developed than other islands in Indonesia. Therefore, we suggest future research addresses multiple rural areas from diverse locations in the country. It will also be interesting to see cross-country comparisons on the same set of variables.

**Author Contributions:** Conceptualization, A.R.T.H. and K.O.; methodology, A.R.T.H.; software, A.R.T.H.; validation, K.O., C.P.M.S. and S.H.; formal analysis, A.R.T.H. and C.P.M.S.; investigation, A.R.T.H.; resources, A.R.T.H. and K.O.; data curation, A.R.T.H.; writing—original draft preparation, A.R.T.H. and C.P.M.S.; writing—review and editing, K.O. and S.H.; visualization, C.P.M.S.; supervision, K.O., C.P.M.S. and S.H.; project administration, A.R.T.H. and K.O.; funding acquisition, A.R.T.H., C.P.M.S. and K.O. All authors have read and agreed to the published version of the manuscript.

**Funding:** This research was funded by the Indonesian Endowment Fund for Education (*Lembaga Pengelola Dana Pendidikan*—LPDP) grant no. 201908220215423. The manuscript preparation and APC were funded by the SPIRITS 2022 of Kyoto University.

**Institutional Review Board Statement:** Not applicable.

**Informed Consent Statement:** Not applicable.

**Data Availability Statement:** The data are available by request.

**Acknowledgments:** All authors would like to thank Rizal K. Maulana, Zendy N. Nursaputra, Aris Subagiyo, and Turniningtyas A. Rachmawati for their supports during the field surveys.

**Conflicts of Interest:** The authors declare no conflict of interest. The funders had no role in the design of the study; in the collection, analyses, or interpretation of data; in the writing of the manuscript; or in the decision to publish the results.

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
