# Peer review of "Distance-Dependent Migration Intention of Villagers: Comparative Study of Peri-Urban and Remote Villages in Indonesia"

_admsci, doi:10.3390/admsci12020048_

Round 1
Reviewer 1 Report
The article deals with a very topical issue of rural migration. The problem is also perceived in other parts of the world, including Europe.
Authors used the Mann-Whitney U, ANOVA, and Brown-Forsythe to test three hypotheses.
I think the choice of topic is original, especially if we look at the conjectures and results that have shown that information sources negatively affect migration intentions in peri-urban settings. The results also shown that place attachment and information sources contribute differently depending on the distance to the urban area.
The methodology used (Mann-Whitney U, ANOVA and Brown-Forsythe) also seems appropriate and well-argued. In the findings, they answered the research questions with arguments.
In general, I believe that the article is prepared correctly and contains all the elements that a scientific contribution at this level needs.
Reviewer 2 Report
Comments
This article presents a theoretical reflection on the sociological issues involved in the debate on rurality and environmental regulation in rural communities. Thinking about rural transport implies rescuing old problems of displacement. In all regions of the country, the issue is the same: for those who live in rural areas of cities, private transport becomes necessary, often being the only way to travel to the urban center. The weakened structure goes through the difficulty of implementing sufficient means of public transport to meet demand. The problem is more accentuated due to the low population rates in these areas, which are insufficient for the development of adequate infrastructure.
Migration can be defined as “the movement of the population in space”, this movement being visualized in various ways, according to the perception of each researcher. Migration can visualize the departure of inhabitants while emigration consists of their arrival as immigration. Therefore, there is a difficulty in conceptualizing the migratory process, or the moment as a migratory phenomenon. Migration can be studied considering the time and distance variables.
The attraction to industrialization makes migrants attracted to work in factories or commerce. While an area is attracted to agriculture, immigrants can be encouraged to help with the food shortage in the state. Other immigrants can also move to work in the formal and informal trade.
Information and Communication Technologies offer numerous advantages, e.g., more access to information, reduced costs at work and more connectivity between people. However, digitization is not happening equally across the world. Is this a problem in these two rural villages in Indonesia? There is also an imbalance and the name for this is digital divide? Or is everyone's accessibility to the internet total? It would be relevant to make a reference in the article to the existence or not of this phenomenon.
The context of mobility is enhanced by the rise of mobile digital technologies and wireless connections. However, when it comes to theorizing and discussing this concept, the authors treat it as an essentially urban phenomenon, thus denying the possibility of building this mobile scenario that strives for instant communication and knowledge production in rural spaces. Therefore, this article seeks to address mobility in a smaller dimension that is not restricted to the rural perimeter, but encompasses the plurality of ways of being and living in rural spaces.
Suggestions
The article addresses issues of territorial mobility between two rural and one urban centers on the island of Java, seeking to focus on a social context in possible networks of interaction and construction of infrastructure and public services. The localities protect what the socio-economic circumstances of the inhabitants preserve. The cultural origin of relationships and behaviors depends on living conditions and habits, customs and attachment to the land.
In the case of rural areas, namely those studied in this article, the authors could explore the creation of cooperatives in various economic and industrial sectors and, therefore, with social impact. They are agents that have always been on the sidelines of the agricultural sectors and that, in this way, regulate themselves with the constraints of public services. This possible future work evaluates the regulatory apparatus for the activities of cooperatives and the public regulatory policy applied in East Java province.
The focus of the evaluation should be the rural citizen, the citizen who consumes goods and services, e.g., electricity and drinking water and not excluded from access to what is elementary, such as work. The interpretation of the problem and the study of the regulation process lead to the verification of the existence of a situation of embarrassment. The regulatory body's attitude towards the task that the law imposes on it can create a paradox in which cooperatives can remain outside the regulator's reach, with greater harm to the citizen who lives in their village or rural area.
When rural inhabitants end up obtaining the guarantee of their right to legally registered property, then citizens who live in the area of ​​cooperatives do not have the protection of the State to enforce this right. The work offers an academic proposal for an alternative way out of this institutional impasse, based on the search for balance between the agents involved in the process, namely public decision-makers.
Public infrastructure and services are essential to maintain territorial balance and social cohesion. In particular, the COVID-19 pandemic highlighted the need to guarantee health and sanitation services to the entire population. In addition to protecting people in situations of relative vulnerability, services and infrastructure allow for a better territorial distribution, providing territories beyond cities. This aspect is not included in the investigation of the presented topic. The future goes this way.